# Ownership succession, risk taking and debt financing from the perspective of the institutional environment

**Yanbo Liu[1], Hanzhou Liu[1], Decai Tang[2]\*, Chenxi Yin[1], Lin Kong[1]**

**1** School of Law and Business, Sanjiang University, Nanjing, China, **2** School of Industry and Urban Construction, Hengxing University, Qingdao, China

\* tangdecai2003@163.com

## Abstract

This study examines family firms listed on China's A-share market, using a two-way fixed effects model to explore the impact of ownership succession on debt financing and the moderating role of the institutional environment. The findings indicate that second-generation ownership succession reduces risk-taking, which in turn lowers debt financing. Additionally, the institutional environment mitigates the negative relationship between second-generation ownership succession, risk-taking, and debt financing. Further analysis reveals that factors such as the development of the non-state-owned economy, the maturity of factor markets, and advancements in market intermediaries and legal systems significantly moderate these relationships. This study broadens the research perspective on intergenerational succession in family businesses and offers empirical insights to help family firms adapt to regional institutional differences more effectively.

## 1. Introduction

In 2023, the Central Committee of the Communist Party of China and the State Council issued the "Opinions on Promoting the Development and Growth of the Private Economy", proposing to speed up the creation of a market-oriented, rule of law, and international first-class business environment, improve financing support policies, and establish a market-oriented risk sharing mechanism involving multiple parties, which pointed out the direction for the development of private enterprises. At the same time, as a new force to promote China 's modernization, private enterprises have made great contributions to China 's economy in terms of scientific and technological innovation, employment absorption, import and export trade and emerging industry development. 85.4% of China 's private enterprises are family businesses. These family businesses are at a critical point of the first generation and the second generation of succession in recent years, and the success or failure will determine the future fate of the family and the enterprise. A review of the development of

**Data availability statement:** All relevant data are within the paper and its Supporting information files.

**Funding:** The author(s) received no specific funding for this work.

**Competing interests:** The authors have declared that no competing interests exist.

family-owned businesses in China reveals that credit issues are a key factor affecting their growth and creating significant challenges [1]. Insufficient financing or improper financial structuring not only hampers the current progress of these enterprises but also impedes the succession process, limiting their potential for future growth [2]. In severe cases, this may result in the loss of family control or bankruptcy and liquidation. Due to the long-standing characteristics of China's state-owned financial capital system, Chinese family businesses face a relative scarcity of debt financing channels, with bank loans being a significant source of their debt capital. Simultaneously, approximately three-quarters of Chinese family businesses are undergoing or are on the verge of a generational transition from the founder to the second generation [3]. This raises the question: What specific impact does intergenerational succession have on the debt financing of family businesses?

For family businesses, the aversion to losing control when external funds are needed can lead them to avoid new equity issuance and rely more on debt than non-family businesses [4]. When family businesses enter the period of intergenerational inheritance, the financial knowledge, attitude towards risk, succession experience, inheritance cost and inheritance plans of the business owners are the main factors affecting the level of debt [5]. Most of the previous scholars ' studies have recognized that second-generation involvement will affect debt financing, but the specific concerns and research conclusions are not consistent. The main research conclusions are as follows: The first view is that in comparison to first and third-generation family businesses, second-generation family enterprises tend to be more conservative, risk-averse and inclined towards debt financing [4]. From the perspective of family reputation, the acquisition of management rights by the second generation sends a signal to the outside world that the family plans to hold and operate the enterprise for a long term, which helps them gain reputation and maintain cooperation with stakeholders, ultimately enhancing the long-term debt financing ability of the enterprise [6]. The second view is that second-generation involvement will reduce the level of debt financing. From the perspective of risk avoidance and maintaining social and emotional wealth, second-generation succession will lead to lower debt financing ratios [7]. The third view is that the generational status of family members does not significantly affect a company's liabilities [8]. Furthermore, a section of scholars believe that the relationship between second-generation members and debt financing is influenced by the management positions they hold [5,9]. Reviewing this literature reveals two main insights. Firstly, most previous studies have focused on the involvement of the second generation in management, with little in-depth research on the impact of the inevitable ownership succession in intergenerational transfers on family business debt financing. Secondly, while previous studies agree that generational succession in family businesses affects debt financing, the specific direction and mechanism of this impact remain unclear, suggesting that the relationship between second-generation ownership succession and debt financing is likely influenced by other moderating variables. Most scholars concur that compared to state-owned enterprises, private family businesses, such as those run by families, are more sensitive to changes in the regional institutional environment [10], and their operational

decisions are more likely to be influenced by this environment, however, there has been no research focusing on the specific impact of institutional environment on debt financing during the intergenerational inheritance period of Chinese family businesses.

Based on the agency theory, when family businesses enter the intergenerational inheritance period, the inheritance of second-generation ownership will lead to changes in the trust relationship between business owners and professional managers, small and medium-sized shareholders, and other stakeholders including creditors, and the agency problems will increase. So, what impact will this have on debt financing, and what is the specific impact mechanism? What role does the institutional environment play in it? Clarifying these issues will help gain a deeper understanding of the potential problems during family business generational transitions. It will also facilitate the access to debt financing resources for these businesses, and promote the smooth intergenerational succession of family enterprises.

Therefore, this paper intends to use data from Chinese family-owned listed companies from 2003 to 2023. Based on the agency theory, this study introduces the institutional environment as a moderating variable to empirically investigate the impact of ownership succession in Chinese family businesses on debt financing and its mechanisms. The marginal contributions of this paper are threefold.

Firstly, it delves into the impact mechanism of second-generation ownership succession on debt financing from the perspective of risk-taking, thus expanding the research perspective on intergenerational succession literature in family businesses.

Secondly, it examines in depth the influence of the institutional environment on the relationship between second-generation ownership succession and debt financing. This reveals the operational mechanisms of environmental variables in the intergenerational succession of family businesses, enriching the research outcomes in the field of generational succession.

Thirdly, the paper further explores the specific institutional environmental factors affecting intergenerational succession in family businesses. It provides empirical evidence for family businesses to better adapt to regional institutional environmental differences, aiding them in better-addressing challenges posed by external environments.

## 2. Theoretical analysis and research hypotheses

### 2.1. Second-generation ownership succession and debt financing

The agency theory holds that the agency problem is caused by the inconsistency of the goals between the principal and the agent, as well as information asymmetry. The symmetrical trust between the principal and the agent supports the inheritance and continuation of the family business [11]. In the context of the second-generation ownership inheritance, changes in the ownership structure of enterprises can lead to changes in the agency relationship, which has a restraining effect on debt financing.

Firstly, before the inheritance of second-generation ownership, the first generation dominated corporate decision-making, and the corporate governance structure was relatively stable. The first generation had a high level of trust with professional managers, shareholders of small and medium-sized enterprises, and other stakeholders including creditors, which could effectively alleviate conflicts of interest among all parties. In such scenarios, when there is a need for external funding, family businesses typically prefer debt over equity financing to ensure family control [12]. Due to credit discrimination against family businesses in the capital market, Chinese listed family businesses tend to have a lower proportion of long-term debt and a higher reliance on short-term debt, primarily sourced from short-term loans [13]. At this point, family businesses often already have higher financial risks. Entering the intergenerational inheritance period, due to the difficulty in transferring assets such as implicit capital and social relationships during inheritance, the trust between the second generation and professional managers, shareholders of small and medium-sized enterprises, and other stakeholders including creditors will decrease compared to the first generation. This will lead to increased conflicts of interest within and

outside the enterprise, and the agency costs will continue to rise. Moreover, in the absence of trust, inheritance expansion will cause family businesses to split [11], thereby losing control of the enterprise. In this situation, in order to reduce agency costs and their own inheritance risks, the second generation will be more cautious about debt financing.

Secondly, from the perspective of creditors, the inheritance of second-generation ownership disrupts the original information balance, increases information asymmetry, and leads to a decrease in debt financing provided by creditors to enterprises. Before the inheritance of second-generation ownership, creditors have a relatively stable understanding of the enterprise through long-term information disclosure and business performance. After the inheritance of second-generation ownership, enterprises may experience power structure adjustments and management changes, and the personal abilities, business concepts, and other information of the second-generation have not been fully transmitted to the market. These changes increase the uncertainty of enterprise operations and exacerbate the degree of information asymmetry between creditors and enterprises. In order to reduce their own risks, creditors will choose to reduce their debt financing to the enterprise.

Therefore, this paper proposes Hypothesis 1:

Hypothesis 1: Second-generation ownership succession inhibits corporate debt financing.

## 2.2. Second-generation ownership succession, risk-taking, and debt financing

Before the inheritance of second-generation ownership, the first generation may be more inclined to take risks to achieve rapid expansion and wealth accumulation of the enterprise. When the second generation acquires ownership, family businesses begin to face a series of risk factors such as organizational restructuring, restructuring of social network relationships, and disruption of harmonious relationships among family members, leading to an increase in principal-agent problems between owners and various stakeholders. In this case, for the sake of family reputation and long-term stable inheritance of the enterprise, both the founder and the second generation tend to restrain the enterprise's risk-taking and reduce debt financing.

On the one hand, from the perspective of the first generation, in order to help the second generation integrate into the enterprise and achieve smooth succession, the first generation has the motivation to create a relatively low-risk environment and adopt low-risk business strategies to ensure the stability of the enterprise [14]. Founders who are in the period of intergenerational inheritance will focus a lot of energy on internal governance of the enterprise, often reducing some high-risk investment activities, and the demand for debt financing has decreased accordingly.On the other hand, from the perspective of the second generation, the higher the level of ownership inherited by the second generation, the higher the degree of involvement of the family in the ownership level. The interests of the second generation, the family, and the enterprise are more consistent. Family members, including the second generation, will have a strong dependence on the enterprise, forming a psychological and emotional attachment to the enterprise, viewing the enterprise as a "family" enterprise [15], and are more unwilling to lose family control. However, at the same time, second-generation individuals often lack sufficient industry knowledge, market insight, and management experience, and may face scrutiny and pressure from within the family and stakeholders. In order to maintain their reputation and status in the family and business, and alleviate agency problems, the second generation will be more inclined to avoid risks when making decisions. The willingness of the company to take financial risks will be suppressed, and high-risk debt financing will be reduced.

Accordingly, this paper proposes Hypothesis 2.

Hypothesis 2: Second-generation ownership succession inhibits corporate risk-taking and decreases debt financing.

## 2.3. The moderating role of the institutional environment

The future fate of an organization is influenced not only by the behavior of its members but also by the external environment, which may bring unforeseeable risks. The institutional environment, a crucial constraint in corporate governance, involves various factors such as industry policies, political and legal systems, government governance boundaries and

resource acquisition, significantly impacting corporate decision-making. For family businesses, establishing close connections with the external environment to obtain external resources and capabilities is vital for sustainable growth [16]. Therefore, this paper introduces the institutional environment as a significant environmental variable to analyze its impact on debt financing decisions during generational succession of family businesses.

The institutional environment for organizational survival and development is complex, encompassing multiple layers and dimensions. These institutions establish key values and conceptual frameworks, such as legitimacy and meaning, which impose varying demands on organizations. To navigate these external institutional pressures, organizations must adopt action strategies that best align with their interests. When family businesses make debt financing decisions, they must weigh both economic and non-economic considerations. The institutional environment in which they operate not only influences access to scarce resources needed for economic benefits but also affects the family's socioemotional wealth. Family businesses in China are spread across regions with diverse institutional conditions [17]. Different institutional environments impose different requirements on enterprises. A good institutional environment can not only effectively alleviate the principal-agent conflicts in family businesses during the intergenerational inheritance period, but also provide stable business expectations and reliable guarantees, improve the financing environment of enterprises, and reduce financing risks.

Firstly, when a company is in a period of intergenerational succession, the trust between owners, managers, and other stakeholders changes, good systems can still be relied upon to alleviate the principal-agent problem between them. On the one hand, regions with better institutional environments usually have higher and more effective supervision mechanisms for corporate information disclosure, which can enhance corporate information transparency, improve information asymmetry, and alleviate the principal-agent problem between major shareholders and small and medium-sized shareholders. On the other hand, a good institutional environment can clearly define the responsibilities and behavioral norms of agents through constraint mechanisms, such as using laws to supervise and restrict the encroachment of major shareholders on the interests of small and medium-sized shareholders, alleviate agency problems, and reduce agency costs.

In addition, a good institutional environment can not only protect the legitimate rights and interests of enterprises, but also reduce the uncertainty and risks they face. In such a market environment, the second generation will believe that the consequences of risk-taking are easier to control and bear, and dare to use more debt financing. On the contrary, if a company is located in an area with a relatively poor institutional environment, conflicts of interest between the company and other stakeholders increase, and agency costs increase. Enterprises not only have to deal with and prevent market risks, but also have to spend energy and resources dealing with inefficient government agencies and complex regulations, ultimately making it difficult for family businesses to obtain the important resources needed to cope with financial and operational risks arising from intergenerational inheritance, thereby suppressing the level of investment and financing risk bearing of enterprises and reducing debt financing.

Therefore, this paper proposes Hypotheses 3 and 4:

Hypothesis 3: The institutional environment weakens the inhibitory effect of second-generation ownership succession on risk-taking.

Hypothesis 4: The institutional environment weakens the negative impact of second-generation ownership succession on debt financing.

## 3. Research design

### 3.1. Sample selection and data collection

This study focuses on family businesses listed on the Shanghai and Shenzhen A-share stock exchanges in China from 2003 to 2023. There are two reasons for this choice: firstly, due to data availability, the data for Chinese family businesses comes from the CSMAR database, which has been in use since 2003–2023; The second reason is that most Chinese family businesses were established in the early stages of reform and opening up, and according to the age of

the founders, intergenerational inheritance began after 2000. The Forbes Family Business Survey Report shows that in 2012, the proportion of family listed companies that completed second-generation succession was approximately 7%; and according to data from the China Private Economy Research Association, from 2017 to 2023, more than three-quarters of family businesses will face handover. This proves that the research time interval in this article is reasonable. Following the criteria established by previous scholars, the second generation in this study is defined as the sons, daughters, sons-in-law, or daughters-in-law of the company's founder. When multiple second-generation members are involved in the same company, the analysis focuses on the member with the largest ownership share. Using the 'Family Business Database' in the GTA (Guotai'an) database, this study initially identifies all family-listed companies on the Shanghai and Shenzhen stock exchanges. The following filtering criteria are then applied: (1) Exclude companies that have been marked as ST or *ST during the sample period; (2) Exclude companies from industries with different applicable accounting standards or more strictly regulated ones, specifically those with industry codes starting with 'J', such as financial and insurance companies; (3) Exclude listed family businesses classified as types "1" and "2", focusing only on type "3" family businesses, defined as "multi-person family enterprises"; (4) Exclude companies with severe data omissions.

According to these criteria, a final effective sample of 742 family businesses was obtained, yielding a 21-year unbalanced panel data set with 4217 observations. To ensure the robustness and reliability of the research conclusions and to mitigate the impact of extreme data on the empirical results, all continuous variables are winsorized at the 1st and 99th percentiles.

### 3.2. Variable design

**3.2.1. Dependent variable.** Debt Financing (DEBT): Bank loans encompassing both long-term and short-term loans constitute the primary source of debt financing for family businesses. Considering that most family businesses have zero long-term loans and that debt funds mainly come from short-term loans, this study, following Jiang Teng et al. [18], uses the ratio of short-term loans to total assets as the measure of debt financing.

**3.2.2. Independent variable.** Second-Generation Ownership Succession (SFO): This study primarily focuses on the impact of the second generation's degree of ownership involvement in family businesses on debt financing decisions. Drawing from Jiang Tao et al. [13], it measures second-generation ownership succession using the proportion of ownership acquired by second-generation family members. If multiple second-generation members exist, the one with the highest ownership proportion is selected for study.

**3.2.3. Mediating variable.** Risk-Taking (RISKTAKE): When a company's level of risk-taking is high, the volatility of its stock returns is also higher. Following Shen Haomin et al. [19],this study, measures risk-taking using the standard deviation of the company's industry-adjusted stock returns over five years.

**3.2.4. Moderating variables.** Institutional Environment(MAR): Following the studies of previous scholars [13,20], this study uses the latest overall index score of marketization compiled by Fan Gang et al. (2022) to measure the overall level of the institutional environment.

**3.2.5. Control variables.** Company Characteristics and Governance Mechanisms: These factors can influence a company's financing capability, the study therefore includes the following control variables: Company Size (SCALE), Leverage (LIA), Cash Flow (CASH), Age of the Company Since Listing (AGE), Proportion of Fixed Assets (FIXED), Growth (DEVE), Market Value (MV), Profitability (EAR), CEO and Chairman Duality (DUAL), and Proportion of Family Executives (TOP).

Additionally, the study includes dummy variables for Industry (IND) and Year (YEAR) to control for industry and time effects. Definitions and measurements of all variables are presented in Table 1.

### 3.3. Model construction

**3.3.1. Second-generation ownership succession and debt financing.** First, the study constructs Model (1), where i represents different companies, and t represents different years. The 'controls' represent a set of control variables. If the coefficient of second-generation ownership succession in Model (1) is significantly negative, Hypothesis 1 is supported.

**Table 1. The definition and measurement of variables.**

| Type | Name | Variable code | The Definition and Measurement of Variables |
|---|---|---|---|
| Dependent Variable | Debt Financing | DEBT | Short-term loan/total assets |
| Independent Variable | Second-Generation Ownership Succession | SFO | The highest ownership ratio of the second-generation family members in the enterprise |
| Mediating Variable | Risk-Taking | RISKTAKE | The risks undertaken |
| Moderating Variables | Institutional Environment | MAR | The Overall Marketization Index data from 《Marketization Index of China's Provinces: Neri Report (2022)》 |
| | Government-Market Relationship | GOV | The Government-Market Relationship Index data from 《Marketization Index of China's Provinces: Neri Report (2022)》 |
| | Property Retention Index | PRI | The Property Retention Index data from 《Marketization Index of China's Provinces: Neri Report (2022)》 |
| | Product Market Development Index | PRO | The Product Market Development Index data from 《Marketization Index of China's Provinces: Neri Report (2022)》 |
| | Growth level of essential factor market | RES | The Growth level data of essential factor market from 《Marketization Index of China's Provinces: Neri Report (2022)》 |
| | Market Intermediary Organization Development and Legal Level | LAW | The Market Intermediary Organization Development and Legal Level Index data from 《Marketization Index of China's Provinces: Neri Report (2022)》 |
| Control Variables | Company Size | SCALE | The logarithm of total assets |
| | Leverage | LIA | Total liabilities/total assets |
| | Proportion of Family Executives | TOP | The proportion of family members serving as executives to the total number of executives |
| | Cash Flow | CASH | Net cash flow generated from operating activities/total assets at the end of the year |
| | Age of the Company Since Listing | AGE | Subtract the year of the company's listing from the year of the current year |
| | Proportion of Fixed Assets | FIXED | Fixed assets/Total assets at the end of the year |
| | Growth | DEVE | Revenue growth rate |
| | Market Value | MV | Market value/total assets |
| | Profitability | EAR | Net profit/owner's equity |
| | Chairman Duality | DUAL | If the chairman and general manager are the same person, the value is 1; otherwise, it is 0. |
| | Dummy Variables for Year | YEAR | Dummy Variables for Year |
| | Dummy Variables for Industry | IND | Dummy Variables for Industry |

$$DEBT_{i,t+1} = \lambda_0 + \lambda_1 SFO_{i,t} + \lambda \sum Controls_{i,t} + \varepsilon_{i,t} \tag{1}$$

**3.3.2. Second-generation ownership succession, risk-taking, and debt financing.** To test the mechanism by which second-generation ownership succession reduces risk-taking and inhibits debt financing, the study constructs Model (2) and Model (3). If the coefficient of second-generation ownership succession in Model (2) is significantly negative, it indicates that second-generation ownership succession inhibits corporate risk-taking. If the coefficient of risk-taking in model (3) is also significant, it indicates that second-generation ownership inheritance suppresses corporate debt financing by reducing risk-taking.

$$RISKTAKE_{i,t+1} = \beta_0 + \beta_1 SFO_{i,t} + \beta \sum Controls_{i,t} + \varepsilon_{i,t} \tag{2}$$

$$DEBT_{i,t+1} = k_0 + k_1 SFO_{i,t} + k_2 RISKTAKE_{i,t} + k \sum controls_{i,t} + \varepsilon_{i,t} \tag{3}$$

### 3.3.3. The moderating effect of the institutional environment.

To test the moderating effect of the institutional environment, the study constructs Models (4) and (5), drawing on the research by Wen Zhonglin et al. [21], and incorporates a set of control variables. If the coefficient of second-generation ownership succession in Model (1) is significantly negative, and the coefficient of the interaction term between second-generation ownership succession and the institutional environment in Model (4) is significantly positive, Hypothesis 3 is supported. Similarly, if the coefficient in Model (5) is significantly positive, Hypothesis 4 is supported..

$$RISKTAKE_{i,t+1} = \eta_0 + \eta_1 SFO_{i,t} + \eta_2 MAR_{i,t} + \eta_3 SFO_{i,t} \times MAR_{i,t} + \eta \sum Controls_{i,t} + \varepsilon_{i,t} \tag{4}$$

$$DEBT_{i,t+1} = \alpha_0 + \alpha_1 SFO_{i,t} + \alpha_2 MAR_{i,t} + \alpha_3 SFO_{i,t} \times MAR_{i,t} + \alpha \sum Controls_{i,t} + \varepsilon_{i,t} \tag{5}$$

These models will help in understanding the dynamics between second-generation ownership succession, debt financing, risk-taking and the moderating role of the institutional environment in Chinese family businesses.

## 4. Empirical results and analysis

### 4.1 Descriptive statistics and correlation analysis

Table 2 reports the descriptive statistics of all variables after winsorization and before regression analysis. From Table 2, it can be observed that:

Debt Financing (DEBT): The mean value is 0.08, with a minimum of 0 and a maximum of 0.655, and a standard deviation of 0.092. This suggests that, on average, family businesses have a relatively low level of debt financing, accounting for only 8% of their assets. Some family businesses refrained from using debt financing in certain years.

Second-Generation Ownership Succession (SFO): The mean value is 0.416, ranging from a minimum of 0.026 and a maximum of 1 with a standard deviation is 0.179. This shows a significant variance in the degree of ownership involvement among second-generation members in family businesses. The highest ownership stake reaches up to 100%, indicating absolute controlling rights and substantial influence over business decisions. Meanwhile, the lowest is 2.6%, averaging at 41.6%, suggesting that most second-generation members in family enterprises have considerable say in business decisions.

Table 2. Descriptive statistical analysis of variables.

| Variable | Sample | Mean Value | Standard Deviation | Minimum value | Maximum value |
|---|---|---|---|---|---|
| DEBT | 4217 | 0.080 | 0.092 | 0.000 | 0.655 |
| SFO | 4,217 | 0.416 | 0.179 | 0.026 | 1.000 |
| RISKTAKE | 4,111 | 1.994 | 0.394 | 0.702 | 5.836 |
| MAR | 3,961 | 10.118 | 1.691 | −0.161 | 12.864 |
| SCALE | 3,957 | 21.908 | 1.030 | 19.034 | 27.011 |
| LIA | 4,217 | 0.371 | 0.186 | 0.008 | 1.484 |
| TOP | 3,888 | 0.166 | 0.075 | 0.035 | 0.467 |
| CASH | 4,217 | 0.054 | 0.074 | −0.658 | 0.580 |
| AGE | 4,217 | 7.688 | 6.316 | −0.318 | 31.773 |
| FIXED | 4,217 | 0.209 | 0.129 | 0.000 | 0.769 |
| DEVE | 4,215 | 0.207 | 1.273 | −0.953 | 58.749 |
| MV | 4,217 | 2.093 | 1.848 | 0.000 | 25.328 |
| EAR | 4,211 | 0.052 | 0.417 | −13.476 | 0.781 |
| DUAL | 4,183 | 0.265 | 0.441 | 0.000 | 1.000 |

Risk-Taking (RISKTAKE): The mean value is 1.994, with a minimum of 0.702 and a maximum of 5.836, and a standard deviation of 0.394. This indicates significant variation in the level of risk-taking among family businesses, with a considerable gap between the minimum and maximum values.

Institutional Environment (MAR): The mean value is 10.118, with a minimum of −0.161 and a maximum of 12.864. The standard deviation is 1.691, reflecting substantial variations in the institutional environment across different regions in China.

These statistics provide a foundational understanding of the variables' distribution and variability, essential for interpreting the results of the regression analysis.

Table 3 reports the correlation analysis results among all the research variables. The key observations from the analysis are as follows:

The correlation coefficient between Second-Generation Ownership Succession (SFO) and Debt Financing (DEBT) is −0.151 ($p < 0.01$) indicating a significant negative correlation. This suggests that as the second generation acquires more ownership in the business, there is a higher likelihood of a negative impact on the level of debt financing. This finding provides preliminary support for Hypothesis 1.

The correlation coefficient between Second-Generation Ownership Succession (SFO) and Risk-Taking (RISKTAKE) is 0.098 ($p < 0.01$).

The correlation between the moderating variable, Institutional Environment (MAR), and both Risk-Taking (RISKTAKE) and Debt Financing (DEBT) is not significant. This absence of a significant correlation makes it difficult to discern, based solely on correlation analysis, to determine the impact of the overall level of the institutional environment on the relationship between second-generation ownership succession and debt financing. Further testing is required to clarify this relationship.

These correlation results provide an initial indication of the relationships between key variables but do not imply causation. A more detailed regression analysis is needed to test the hypotheses thoroughly and understand the underlying mechanisms.

**Table 3. Variable correlation analysis.**

| Variable | DEBT | SFO | RISKTAKE | MAR | SCALE | TOP | CASH | AGE | ALR | FIXED | DEVE | MV | EAR |
|---|---|---|---|---|---|---|---|---|---|---|---|---|---|
| DEBT | 1 | | | | | | | | | | | | |
| SFO | −0.151*** | 1 | | | | | | | | | | | |
| RISKTAKE | −0.073*** | 0.098*** | 1 | | | | | | | | | | |
| MAR | −0.006 | 0.105*** | 0.022 | 1 | | | | | | | | | |
| SCALE | 0.214*** | −0.193*** | −0.262*** | 0.061*** | 1 | | | | | | | | |
| TOP | 0.529*** | −0.140*** | −0.090*** | 0.012 | 0.486*** | 1 | | | | | | | |
| CASH | −0.044*** | 0.270*** | 0.059*** | 0.085*** | −0.194*** | −0.177*** | 1 | | | | | | |
| AGE | −0.204*** | 0.073*** | −0.016 | 0.013 | 0.039** | −0.145*** | 0.033** | 1 | | | | | |
| LIA | 0.137*** | −0.412*** | −0.264*** | −0.086*** | 0.472*** | 0.280*** | −0.282*** | −0.032** | 1 | | | | |
| FIXED | 0.177*** | −0.063*** | −0.056*** | −0.135*** | 0.075*** | 0.071*** | 0.040** | 0.202*** | 0.018 | 1 | | | |
| DEVE | −0.007 | −0.007 | 0.032** | −0.015 | 0.037** | 0.040*** | −0.022 | −0.022 | 0.038** | −0.005 | 1 | | |
| MV | −0.236*** | 0.130*** | 0.373*** | −0.040** | −0.355*** | −0.394*** | 0.040** | −0.022 | −0.185*** | −0.105*** | 0.018 | 1 | |
| EAR | −0.129*** | 0.048*** | −0.007 | −0.007 | −0.010 | −0.164*** | 0.032** | −0.022 | −0.052*** | 0.005 | 0.043*** | 0.061*** | 1 |
| DUAL | 0.047*** | 0.050*** | 0.051*** | 0.098*** | −0.035* | −0.008 | −0.140*** | −0.022 | −0.01 | −0.062*** | −0.001 | 0.086*** | −0.006 |

Note: The numbers in the table represent the correlation coefficients between pairs of variables. An asterisk (*) denotes significance at the 10% level, double asterisks (**) denote significance at the 5% level, and triple asterisks (***) indicate significance at the 1% level.

## 4.2. Multicollinearity test

Table 4 presents the results of the multicollinearity test for various variables. The highest Variance Inflation Factor (VIF) value in the model is 1.75, and all VIF values are less than 2, which is well below the threshold of 10. This indicates that the research model does not suffer from severe multicollinearity.

## 4.3. Regression model selection

Before proceeding with the regression analysis, an F-test was conducted to select the appropriate model. The F-test result, with an F-value of 7.55(p<0.01), led to the rejection of the mixed model. Subsequently, a Hausman test was performed, indicating a chi-squared (12) value of 98.51 (p<0.01), leading to the exclusion of the random effects model. Therefore, a fixed effects model was chosen for the analysis.

Additionally, in addressing potential issues of heteroskedasticity, time-series and cross-sectional autocorrelation, and to enhance the robustness of the regression results, robust standard errors were used to adjust all regression models. Considering the lagging effect of independent variables on dependent variables and to mitigate endogeneity issues stemming from reverse causality, the lagged values of the dependent variables were included in the regression model. Furthermore, industry and year controls were also included in all models, and a two-way fixed effects model was applied to further alleviate endogeneity concerns.

This approach contributes to ensuring the reliability and validity of the empirical analysis, providing a more accurate understanding of the relationships between second-generation ownership succession, debt financing, risk-taking, and the institutional environment in the context of Chinese family businesses.

## 4.4. Analysis of regression results

### 4.4.1. Regression results.
Table 5 presents the regression results of the relationships between second-generation ownership succession and debt financing. Column (1) indicates the regression results with only the explanatory variables included in the model, showing a coefficient for second-generation ownership of −0.037 (P<0.05). Column (2) includes both the explanatory variables and all control variables, including year, with the coefficient for second-generation ownership recorded as −0.035 (P<0.05). Column (3) incorporates explanatory variables, all control variables, and employs regression using robust standard errors clustered at the company level. The coefficient of second-generation ownership is −0.048 (P<0.05). This implies that higher levels of second-generation ownership succession are associated with lower levels of family business debt financing. Thus, for every unit increase in second-generation ownership succession, debt financing decreases by 0.048 units. Hypothesis 1 is validated, indicating that second-generation involvement in ownership does adversely affect the debt financing of family businesses.

### 4.4.2. Endogenous concerns.

(1) Propensity Score Matching Method

To address the issue of sample selectivity bias, this study employs the propensity score matching (PSM) method to test robustness. First, the samples were divided into high and low groups based on the median of second-generation ownership, with dummy variables assigned accordingly. Next, all control variables from the previous analysis were used as covariates for propensity score matching. Finally, regression tests were conducted on the matched samples. The results

**Table 4. Variance inflation factor test results.**

| Variable | SFO | RISKTAKE | MAR | DUAL | FIXED | LIA | AGE |
|----------|-----|----------|-----|------|-------|-----|-----|
| VIF | 1.25 | 1.24 | 1.07 | 1.05 | 1.10 | 1.57 | 1.57 |
| Variable | CASH | TOP | MV | EAR | DEVE | SCALE | Mean Value |
| VIF | 1.14 | 1.18 | 1.47 | 1.05 | 1.01 | 1.75 | 1.27 |

**Table 5. Regression analysis results.**

| Variable | (1) | (2) | (3) |
|---|---|---|---|
| | DEBT | DEBT | DEBT |
| SFO | −0.037** | −0.035** | −0.048** |
| | (0.015) | (0.016) | (0.022) |
| SCALE | | 0.002 | 0.001 |
| | | (0.003) | (0.006) |
| LIA | | 0.154*** | 0.154*** |
| | | (0.012) | (0.018) |
| TOP | | 0.151*** | 0.154*** |
| | | (0.034) | (0.059) |
| CASH | | −0.094*** | −0.090** |
| | | (0.016) | (0.035) |
| AGE | | −0.006 | −0.006* |
| | | (0.005) | (0.003) |
| FIXED | | 0.032** | 0.020 |
| | | (0.016) | (0.021) |
| DEVE | | 0.0004 | 0.0004 |
| | | (0.001) | (0.001) |
| MV | | −0.001 | −0.001 |
| | | (0.001) | (0.001) |
| EAR | | −0.022*** | −0.022*** |
| | | (0.004) | (0.008) |
| DUAL | | 0.003 | 0.004 |
| | | (0.004) | (0.006) |
| YEAR | NO | YES | YES |
| IND | NO | NO | YES |
| FIRM | NO | NO | YES |
| Constant | 0.100*** | 0.038 | 0.087 |
| | (0.007) | (0.088) | (0.131) |
| Observations | 3,413 | 3,325 | 3,325 |
| R-squared | 0.002 | 0.204 | 0.239 |

Note: The numbers in the table represent the correlation coefficients between pairs of variables. An asterisk (*) *denotes significance at the 10% level, double asterisks (*\**) **denote significance at the 5% level, and triple asterisks (\*\*\*)** indicate significance at the 1% level.

of one-to-one and one-to-four matching are presented in Columns (1) and (2) of Table 6. The findings indicate that second-generation ownership significantly inhibits debt financing, providing further support for Hypothesis 1.

(2) Instrumental variable method

To further address endogeneity concerns arising from reverse causality, this study uses the age of the first-generation founder (FOUNDER) as an instrumental variable. The rationale is that as the first-generation founder's age increases, the likelihood of the second generation entering the enterprise and obtaining ownership also increases. However, the first-generation founder's age has no significant direct impact on corporate debt financing. Thus, it is appropriate to use the first generation founder's age as an instrumental variable. The results of the first-stage regression, presented in Column (1) of Table 7, show a coefficient of 0.005 (P < 0.01), indicating a significant positive correlation between the instrumental variable and the explanatory variable. The Cragg-Donald Wald F-statistic is 10.02, and the Chi-square value of the

**Table 6. Results of propensity score matching method.**

| Variable | (1)<br>DEBT<br>one-to-one matches | (2)<br>DEBT<br>one-to-four matches |
|---|---|---|
| SFO | −0.075*** | −0.054*** |
|  | (0.016) | (0.012) |
| SCALE | −0.007** | −0.001 |
|  | (0.003) | (0.003) |
| LIA | 0.216*** | 0.194*** |
|  | (0.016) | (0.011) |
| TOP | 0.123*** | 0.120*** |
|  | (0.037) | (0.028) |
| CASH | −0.138*** | −0.124*** |
|  | (0.025) | (0.017) |
| AGE | 0.0002 | 0.0002 |
|  | (0.001) | (0.0005) |
| FIXED | 0.057*** | 0.047*** |
|  | (0.019) | (0.014) |
| DEVE | 0.0001 | −0.0002 |
|  | (0.001) | (0.001) |
| MV | −0.001 | −0.002** |
|  | (0.001) | (0.001) |
| EAR | −0.013 | −0.038*** |
|  | (0.016) | (0.007) |
| DUAL | 0.017*** | 0.011*** |
|  | (0.005) | (0.004) |
| YEAR | YES | YES |
| IND | YES | YES |
| FIRM | YES | YES |
| Constant | 0.321*** | 0.123* |
|  | (0.104) | (0.073) |
| Observations | 1,517 | 2,760 |
| Within $R^2$ | 0.215 | 0.232 |

Note: The numbers in the table represent the correlation coefficients between pairs of variables. An asterisk (*) *denotes significance at the 10% level,* *double asterisks (***)* **denote significance at the 5% level, and triple asterisks (***)** indicate significance at the 1% level.

Stock-Right LM S statistic is 6.55, indicating no issues with weak instrumental variables or overidentification. Column (2) of Table 7 presents the results of the second-stage regression, where the coefficient for second-generation ownership is −0.658 ($P < 0.05$). This confirms that Hypothesis 1 remains valid even after accounting for the reverse causal relationship between second-generation ownership and debt financing..

## 5. Robustness tests

### 5.1. Sample transformation method

The previous research period spanned from 2003 to 2023. However, due to the small sample size from 2003 to 2008 and the influence of the 2008 financial crisis on debt financing in 2009, the sample from 2010 to 2023 was

**Table 7. Results of instrumental variable method.**

| Variable | (1) | (2) |
|---|---|---|
| | SFO | DEBT |
| | First stage | Second stage |
| FOUNDER | 0.005*** | |
| | (0.002) | |
| SFO | | −0.658** |
| | | (0.322) |
| SCALE | −0.035*** | −0.019 |
| | (0.004) | (0.012) |
| LIA | 0.077*** | 0.203*** |
| | (0.014) | (0.029) |
| TOP | 0.221*** | 0.283*** |
| | (0.041) | (0.082) |
| CASH | 0.023 | −0.097*** |
| | (0.019) | (0.021) |
| AGE | −0.021*** | −0.016** |
| | (0.006) | (0.008) |
| FIXED | −0.038* | 0.001 |
| | (0.019) | (0.024) |
| DEVE | 0.002* | 0.001 |
| | (0.001) | (0.001) |
| MV | −0.001 | −0.001 |
| | (0.001) | (0.001) |
| EAR | 0.007 | −0.020*** |
| | (0.004) | (0.005) |
| DUAL | 0.024 | 0.019** |
| | (0.005) | (0.009) |
| YEAR | YES | YES |
| IND | YES | YES |
| FIRM | YES | YES |
| Observations | 3176 | 3,176 |
| R-squared | 0.291 | −0.184 |

Note: The numbers in the table represent the correlation coefficients between pairs of variables. An asterisk (*) *denotes significance at the 10% level,* *double asterisks (***) **denote significance at the 5% level, and triple asterisks (***) indicate significance at the 1% level.

used for the robustness test. The regression results, presented in Column (1) of Table 8, show a coefficient for second-generation ownership of −0.041 ($P < 0.1$), indicating that second-generation ownership significantly inhibits debt financing.

## 5.2. Substitution variable method

To address potential measurement errors in variables, new measurement methods are used to assess the levels of debt financing. The level of debt financing is re-measured using the ratio of short-term loans to total liabilities(DEBT1). The results of the tests with these replacement variables are presented in column (2) of Table 8, and the regression results once again validate the hypothesis 1 proposed in the study.

**Table 8. Results of robustness tests.**

| Variable | (1) | (2) | (3) |
|---|---|---|---|
| | DEBT | DEBT1 | DEBT |
| SFO | −0.041* | −0.094* | −0.048** |
| | (0.023) | (0.053) | (0.022) |
| SCALE | 0.0004 | 0.005 | 0.002 |
| | (0.006) | (0.013) | (0.006) |
| LIA | 0.153*** | 0.129*** | 0.157*** |
| | (0.018) | (0.049) | (0.018) |
| TOP | 0.156** | 0.339** | 0.144** |
| | (0.062) | (0.145) | (0.058) |
| CASH | −0.070** | −0.168** | −0.089*** |
| | (0.031) | (0.069) | (0.035) |
| AGE | −0.006* | −0.009 | −0.005 |
| | (0.003) | (0.008) | (0.003) |
| FIXED | 0.018 | 0.058 | 0.019 |
| | (0.023) | (0.048) | (0.021) |
| DEVE | −0.001** | 0.002 | 0.0004 |
| | (0.001) | (0.001) | (0.001) |
| MV | −0.001 | −0.005* | |
| | (0.001) | (0.002) | |
| EAR | −0.022*** | 0.003 | −0.022*** |
| | (0.008) | (0.005) | (0.008) |
| DUAL | 0.001 | 0.008 | |
| | (0.006) | (0.014) | |
| YEAR | YES | YES | YES |
| IND | YES | YES | YES |
| FIRM | YES | YES | YES |
| Constant | 0.092 | 0.139 | 0.068 |
| | (0.141) | (0.293) | (0.127) |
| Observations | 3,203 | 3,325 | 3,350 |
| R-squared | 0.168 | 0.166 | 0.237 |

Note: The numbers in the table represent the correlation coefficients between pairs of variables. An asterisk (*) denotes significance at the 10% level, double asterisks (**) denote significance at the 5% level, and triple asterisks (***) indicate significance at the 1% level.

## 5.3. Reducing the number of control variables method

To further assess the robustness of the model, two control variables—Market Value (MV) and CEO-Chairman Duality (DUAL)—were excluded from the regression model. The regression was then re-estimated using a two-way fixed effects model, controlling for industry and year. The results, presented in Column (3) of Table 8, remain consistent with the earlier findings.

## 6. Mediating effect analysis of risk-taking

To examine the mediating role of risk-taking, the study first tests the impact of second-generation ownership on reducing risk-taking, based on Model (2). The results in Column (1) show that the regression coefficient for second-generation ownership is −0.085 (P < 0.01), indicating a significant reduction in risk-taking. Furthermore, in Model

(3), risk-taking is included as a mediating variable in the regression model. The outcomes in Column (2) reveal a regression coefficient for risk-taking of 0.042 (P < 0.1). These findings, along with previous results, support Hypothesis 2, demonstrating that second-generation ownership inhibits corporate debt financing by reducing risk-taking [22–23].

Additionally, the study employs the bootstrap method to further test the mediating effect. The confidence interval for the indirect effect product coefficient is [−0.0039774, −0.0007814], excluding zero. This confirms that risk-taking mediates the relationship between second-generation ownership and debt financing, providing further support for Hypothesis 2.

## 7. Moderating effect analysis of institutional environment

Column (3) of Table 9 examines the moderating effect of the institutional environment on the relationship between second-generation ownership inheritance and risk-taking. The results show that the coefficient for the interaction term between second-generation ownership and the institutional environment is 0.022 (P < 0.1), indicating that the institutional environment weakens the inhibitory effect of second-generation ownership inheritance on corporate risk-taking, thus supporting Hypothesis 3.

Column (4) of Table 9 explores the moderating effect of the institutional environment on the relationship between second-generation ownership inheritance and debt financing. The coefficient for the interaction term is 0.021 (P < 0.05), suggesting that the institutional environment similarly weakens the inhibitory effect of second-generation ownership inheritance on debt financing. These findings validate Hypothesis 4.

## 8. Further research: the impact of specific elements of the institutional environment

Considering China's ongoing transition from a government-led planned economy to a market economy, which began in the eastern coastal re\gions and gradually extended to the central and western inland areas, there are significant regional gradients in the transformation process. This disparity is evident not only in the overall level of the institutional environment but also in specific environmental factors, which influence debt financing during family intergenerational succession.

In this context, the study introduces five moderating variables to further explore the role of sub-indicators in institutional environment moderating, the relationship between second-generation ownership succession, risk-taking, and debt financing. These variables are:

(1) Government-Market Relationship (GOV)

(2) Non-State Economy Development Level (PRI)

(3) Product Market Development Level (PRO)

(4) Factor Market Development Level (RES)

(5) Market Intermediary Organizations and Legal Development Level (LAW)

Table 10 displays the regression results of the moderating effects of these five detailed environmental indicators on the relationship between second-generation ownership succession, risk-taking, and debt financing. The results indicate that the development levels of the non-state economy, factor markets, market intermediary organizations, and legal systems significantly weaken the inhibitory effect of second-generation ownership succession on risk-taking. Additionally, the development levels of the non-state economy, factor markets, market intermediary organizations, and legal systems significantly weaken the inhibitory effect of second-generation ownership succession on debt financing.

This suggests that higher levels of non-state economic development, factor market development, market intermediary organizations, and legal systems typically indicate that the economic activities of non-state economies are less restricted. Businesses in turn receive fairer treatment, have access to more economic resources, enjoy a more advanced financial development level, a more comprehensive financial system, higher competition in the financial industry, and witness a

**Table 9. Mediating effect of risk-taking and moderating effect of institutional environment.**

| Variable | (1) RISK Model(2) | (2) DEBT Model(3) | (3) RISK Model(4) | (4) DEBT Model(5) |
|---|---|---|---|---|
| SFO | −0.085*** | −0.045** | −0.070** | −0.035 |
| | (0.030) | (0.022) | (0.030) | (0.022) |
| RISK | | 0.042* | | |
| | | (0.025) | | |
| MAR | | | 0.006 | 0.004 |
| | | | (0.004) | (0.003) |
| SFO*MAR | | | 0.022* | 0.021** |
| | | | (0.012) | (0.009) |
| SCALE | 0.013 | 0.001 | 0.014* | 0.002 |
| | (0.008) | (0.006) | (0.008) | (0.006) |
| LIA | 0.024 | 0.151*** | 0.019 | 0.150*** |
| | (0.024) | (0.018) | (0.024) | (0.018) |
| TOP | −0.154*** | 0.162*** | −0.152*** | 0.155*** |
| | (0.059) | (0.060) | (0.059) | (0.059) |
| CASH | 0.031 | −0.091*** | 0.028 | −0.093*** |
| | (0.025) | (0.035) | (0.025) | (0.035) |
| AGE | −0.039*** | −0.005 | −0.041*** | −0.007** |
| | (0.003) | (0.003) | (0.004) | (0.003) |
| FIXED | −0.061* | 0.021 | −0.062* | 0.019 |
| | (0.033) | (0.021) | (0.033) | (0.021) |
| DEVE | 0.0002 | −0.0004 | 0.0001 | −0.0004 |
| | (0.001) | (0.001) | (0.001) | (0.001) |
| MV | 0.001 | −0.001 | 0.001 | −0.001 |
| | (0.001) | (0.001) | (0.001) | (0.001) |
| EAR | −0.017 | −0.021*** | −0.016 | −0.022*** |
| | (0.011) | (0.007) | (0.011) | (0.008) |
| DUAL | 0.007 | 0.004 | 0.008 | 0.005 |
| | (0.007) | (0.006) | (0.007) | (0.006) |
| YEAR | YES | YES | YES | YES |
| IND | YES | YES | YES | YES |
| FIRM | YES | YES | YES | YES |
| Constant | −0.010 | 0.074 | −0.081 | 0.028 |
| | (0.174) | (0.132) | (0.181) | (0.128) |
| Observations | 3,324 | 3,325 | 3,324 | 3,325 |
| R-squared | 0.133 | 0.241 | 0.136 | 0.242 |

Note: The numbers in the table represent the correlation coefficients between pairs of variables. An asterisk (*) denotes significance at the 10% level, double asterisks (**) denote significance at the 5% level, and triple asterisks (***) indicate significance at the 1% level.

more market-oriented credit funds distribution. Moreover, the judicial department's protection of the rights of production enterprises, consumers, and intellectual property rights becomes stronger. These institutional safeguards increase the willingness of family businesses to take risks, weakening the negative relationship between second-generation ownership succession and debt financing.

 

**Table 10. Further research: the impact of specific elements of the institutional environment.**

| Variable | (1) | (2) | (3) | (4) | (5) | (6) | (7) | (8) | (9) | (10) |
|---|---|---|---|---|---|---|---|---|---|---|
| | RISK | DEBT | RISK | DEBT | RISK | DEBT | RISK | DEBT | RISK | DEBT |
| SFO | −0.078*** | −0.041* | −0.078** | −0.041* | −0.072** | −0.036 | −0.070** | −0.032 | −0.060** | −0.027 |
| | (0.030) | (0.022) | (0.030) | (0.022) | (0.029) | (0.022) | (0.030) | (0.023) | (0.029) | (0.023) |
| GOV | 0.001 | −0.003 | | | | | | | | |
| | (0.002) | (0.002) | | | | | | | | |
| SFO*GOV | −0.012 | −0.012 | | | | | | | | |
| | (0.009) | (0.008) | | | | | | | | |
| PRI | | | 0.005 | 0.001 | | | | | | |
| | | | (0.003) | (0.004) | | | | | | |
| SFO*PRI | | | 0.019* | 0.018** | | | | | | |
| | | | (0.010) | (0.008) | | | | | | |
| PRO | | | | | −0.002 | 0.002 | | | | |
| | | | | | (0.003) | (0.002) | | | | |
| SFO*PRO | | | | | −0.018** | −0.014** | | | | |
| | | | | | (0.008) | (0.007) | | | | |
| RES | | | | | | | 0.001 | 0.002 | | |
| | | | | | | | (0.002) | (0.002) | | |
| SFO*RES | | | | | | | 0.008* | 0.008** | | |
| | | | | | | | (0.005) | (0.004) | | |
| LAW | | | | | | | | | 0.003* | 0.0004 |
| | | | | | | | | | (0.002) | (0.002) |
| SFO*LAW | | | | | | | | | 0.011** | 0.009*** |
| | | | | | | | | | (0.004) | (0.003) |
| SCALE | 0.012 | 0.0001 | 0.013 | 0.001 | 0.012 | 0.0002 | 0.013 | 0.001 | 0.014 | 0.002 |
| | (0.008) | (0.006) | (0.008) | (0.006) | (0.008) | (0.006) | (0.008) | (0.006) | (0.008) | (0.006) |
| LIA | 0.023 | 0.154*** | 0.022 | 0.152*** | 0.022 | 0.152*** | 0.021 | 0.152*** | 0.017 | 0.150*** |
| | (0.024) | (0.018) | (0.024) | (0.018) | (0.024) | (0.018) | (0.024) | (0.018) | (0.024) | (0.018) |
| TOP | −0.153*** | 0.154*** | −0.152** | 0.153*** | −0.156*** | 0.148** | −0.154*** | 0.154*** | −0.151** | 0.154*** |
| | (0.059) | (0.059) | (0.059) | (0.059) | (0.059) | (0.059) | (0.059) | (0.059) | (0.059) | (0.059) |
| CASH | 0.030 | −0.090** | 0.029 | −0.092*** | 0.031 | −0.091*** | 0.029 | −0.092*** | 0.028 | −0.092*** |
| | (0.025) | (0.035) | (0.025) | (0.035) | (0.025) | (0.035) | (0.025) | (0.035) | (0.025) | (0.035) |
| AGE | −0.039*** | −0.005 | −0.041*** | −0.006 | −0.038*** | −0.006 | −0.039*** | −0.008** | −0.043*** | −0.006 |
| | (0.003) | (0.003) | (0.004) | (0.004) | (0.003) | (0.003) | (0.004) | (0.004) | (0.005) | (0.004) |
| FIXED | −0.064* | 0.018 | −0.060* | 0.02 | −0.061* | 0.019 | −0.063* | 0.018 | −0.062* | 0.018 |
| | (0.033) | (0.022) | (0.032) | (0.021) | (0.032) | (0.021) | (0.033) | (0.021) | (0.033) | (0.021) |
| DEVE | 0.0002 | 0.0004 | 0.0001 | 0.0004 | 0.0004 | −0.0004 | 0.0001 | −0.0003 | 0.0001 | −0.0004 |
| | (0.001) | (0.001) | (0.001) | (0.001) | (0.001) | (0.001) | (0.001) | (0.001) | (0.001) | (0.001) |
| MV | 0.001 | −0.001 | 0.001 | −0.001 | 0.001 | −0.001 | 0.001 | −0.001 | 0.001 | −0.001 |
| | (0.001) | (0.001) | (0.001) | (0.001) | (0.001) | (0.001) | (0.001) | (0.001) | (0.001) | (0.001) |
| EAR | −0.017 | −0.022*** | −0.016 | −0.022*** | −0.016 | −0.022*** | −0.017 | −0.022*** | −0.016 | −0.022*** |
| | (0.011) | (0.008) | (0.011) | (0.008) | (0.011) | (0.008) | (0.011) | (0.008) | (0.011) | (0.008) |
| DUAL | 0.007 | 0.004 | 0.008 | 0.005 | 0.008 | 0.005 | 0.008 | 0.005 | 0.008 | 0.005 |
| | (0.007) | (0.006) | (0.007) | (0.006) | (0.007) | (0.006) | (0.007) | (0.006) | (0.007) | (0.006) |
| YEAR | YES | YES | YES | YES | YES | YES | YES | YES | YES | YES |
| IND | YES | YES | YES | YES | YES | YES | YES | YES | YES | YES |
| FIRM | YES | YES | YES | YES | YES | YES | YES | YES | YES | YES |

*(Continued)*

**Table 10.** (Continued)

| Variable | (1) | (2) | (3) | (4) | (5) | (6) | (7) | (8) | (9) | (10) |
|---|---|---|---|---|---|---|---|---|---|---|
| | RISK | DEBT | RISK | DEBT | RISK | DEBT | RISK | DEBT | RISK | DEBT |
| Constant | −0.009 | 0.117 | −0.06 | 0.069 | 0.008 | 0.085 | −0.025 | 0.067 | −0.03 | 0.064 |
| | (0.179) | (0.131) | (0.180) | (0.134) | (0.171) | (0.131) | (0.173) | (0.127) | (0.170) | (0.130) |
| Observations | 3,324 | 3,325 | 3,324 | 3,325 | 3,324 | 3,325 | 3,324 | 3,325 | 3,324 | 3,325 |
| R-squared | 0.134 | 0.241 | 0.136 | 0.241 | 0.135 | 0.241 | 0.135 | 0.243 | 0.139 | 0.243 |

Note: The numbers in the table represent the correlation coefficients between pairs of variables. An asterisk (*) *denotes significance at the 10% level,* *double asterisks (**) **denote significance at the 5% level, and triple asterisks (***)** indicate significance at the 1% level.*

From Table 10, it is also observed that the government-market relationship has no significant impact on the relationship between the inheritance of second-generation ownership and risk-taking, as well as debt financing. This may be attributed to the fact that a better government-market relationship is less likely to lead to arbitrary interventions in capital market resource allocation, thus having a relatively smaller impact on corporate debt financing, consistent with previous studies [24].

Moreover, the environmental factor of product market development not only significantly strengthened the inhibitory effect of second-generation ownership inheritance on risk-taking but also enhanced its inhibitory effect on debt financing. A possible explanation is that the product market development index reflects the degree of local protectionism in different regions, measured by the trade protection measures encountered by enterprises when selling products. A higher index value indicates fewer market access restrictions and more intense competition. This heightened competition may increase the likelihood of intergenerational inheritance failure in family businesses, reduce the second generation's willingness to take risks, and ultimately limit access to debt resources.

## 9. Research conclusions and implications

Intergenerational inheritance is the key aspect of the survival and development of family businesses, and debt financing is an important decision for enterprises. However, the impact of intergenerational succession on debt financing is far from consistent, and the important factor of institutional environment has not been included in the research framework [6–7]. This article incorporates the institutional environment into the research framework and analyzes the impact of second-generation ownership inheritance on debt financing. From the perspective of agency theory, it points out that second-generation ownership inheritance will have an impact on corresponding agency problems, which in turn will affect corporate debt financing. The institutional environment has a weakening effect on the negative relationship between the inheritance of second-generation ownership and debt financing. Specifically, after the second-generation inheritance obtains higher ownership, the agency relationship between the owner and professional managers, small and medium-sized shareholders, and other stakeholders changes, leading to a continuous increase in agency costs. In order to maintain family interests and reduce agency costs, the second generation tends to reduce risk-taking and debt financing. A good institutional environment has put forward higher requirements for information disclosure for enterprises. Through supervision and constraint mechanisms, conflicts of interest between enterprises and professional managers, small and medium-sized shareholders, and other stakeholders have been alleviated, providing a stable operating environment and guarantee for enterprises, weakening the inhibitory effect of second-generation ownership inheritance on debt financing, and explaining the regulatory effect of the institutional environment. The research conclusion of this article has made certain theoretical contributions and management practice inspirations to existing research.

Firstly, This article contributes to the theoretical research on the involvement of second-generation family businesses. The existing research conclusions on the relationship between second-generation involvement and debt

financing are not consistent [6–7]. This article is based on agency theory and confirms the research conclusion of Yao Qingtie and Guo Ping (2012) that the agency chain of family businesses in intergenerational inheritance will change due to trust relationships. The involvement of second-generation ownership will lead to an increase in agency costs between business owners and stakeholders due to a decrease in trust levels, which in turn will affect debt financing.

Secondly, this article has made a contribution to the theoretical research on debt financing of family businesses. This article focuses on the debt financing decisions of family businesses at the important point of intergenerational inheritance, and finds that the involvement of second-generation ownership inhibits corporate debt financing, confirming the previous scholars' view that second-generation involvement can lead to changes in the governance mechanism of family businesses, thereby affecting debt financing [5]. Moreover, this article combines the "Chinese perspective" to conduct research on environmental factors with "Chinese characteristics"[25], exploring the impact of institutional environment on intergenerational debt financing. It is believed that a good institutional environment can alleviate the principal-agent problem between the second generation and stakeholders, create relaxed and fair competition opportunities for family businesses, and weaken the inhibitory effect of second-generation ownership involvement on debt financing.

Thirdly, this article has important practical guidance significance for debt financing decisions of Chinese family businesses in the intergenerational inheritance period. When family businesses enter the intergenerational inheritance period, appropriate debt financing decisions should be made based on external environmental conditions and the company's own risk bearing situation. Specifically, if a family business is located in an area with a poor institutional environment, a succession plan should be developed in advance. It is necessary for the first generation to help the second generation establish trust relationships with other stakeholders, alleviate agency conflicts, and appropriately reduce debt financing to reduce risks.

## Supporting information

**S1 Data. Data from Chinese family-owned listed companies from 2003 to 2023.**
(XLSX)

## Author contributions

**Conceptualization:** Yanbo Liu, Decai Tang.

**Data curation:** Yanbo Liu, Decai Tang.

**Formal analysis:** Yanbo Liu, Hanzhou Liu, Decai Tang.

**Investigation:** Hanzhou Liu, Chenxi Yin, Lin Kong.

**Methodology:** Yanbo Liu, Hanzhou Liu.

**Project administration:** Decai Tang.

**Resources:** Yanbo Liu.

**Software:** Yanbo Liu, Hanzhou Liu.

**Supervision:** Decai Tang.

**Validation:** Yanbo Liu, Hanzhou Liu, Decai Tang, Chenxi Yin.

**Visualization:** Decai Tang, Lin Kong.

**Writing – original draft:** Yanbo Liu, Hanzhou Liu, Decai Tang.

**Writing – review & editing:** Yanbo Liu, Decai Tang, Chenxi Yin, Lin Kong.

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
