## [Decision Letter · Decision Letter 0]

PONE-D-24-11863Ownership Succession, Risk Taking and Debt Financing from the Perspective of the Institutional EnvironmentPLOS ONE

Dear Dr. Tang,

Thank you for submitting your manuscript to PLOS ONE. After careful consideration, we feel that it has merit but does not fully meet PLOS ONE’s publication criteria as it currently stands. Therefore, we invite you to submit a revised version of the manuscript that addresses the points raised during the review process.

We look forward to receiving your revised manuscript.

Kind regards,

Gianluca Mattarocci, PhD

Academic Editor

PLOS ONE

Journal Requirements: When submitting your revision, we need you to address these additional requirements. 1. Please ensure that your manuscript meets PLOS ONE's style requirements, including those for file naming. The PLOS ONE style templates can be found at https://journals.plos.org/plosone/s/file?id=wjVg/PLOSOne_formatting_sample_main_body.pdf and https://journals.plos.org/plosone/s/file?id=ba62/PLOSOne_formatting_sample_title_authors_affiliations.pdf

Reviewers' comments:

Reviewer's Responses to Questions

**Comments to the Author**

1. Is the manuscript technically sound, and do the data support the conclusions?

Reviewer #1: Yes

2. Has the statistical analysis been performed appropriately and rigorously? 

Reviewer #1: Yes

3. Have the authors made all data underlying the findings in their manuscript fully available?

Reviewer #1: Yes

4. Is the manuscript presented in an intelligible fashion and written in standard English?

Reviewer #1: Yes

5. Review Comments to the Author

Reviewer #1: This paper examines the impact of ownership succession on family firms' debt financing and the mechanisms through which the institutional environment influences this relationship using a sample of Chinese family-controlled listed firms. The authors find that second-generation ownership succession tends to inhibit risk-taking, leading to a reduction in debt financing for the enterprise.

6. PLOS authors have the option to publish the peer review history of their article (what does this mean? ). If published, this will include your full peer review and any attached files.

**Do you want your identity to be public for this peer review?** For information about this choice, including consent withdrawal, please see our Privacy Policy .

Reviewer #1: **Yes: ** Baili Yang

---

## [Author Response · Author response to Decision Letter 1]

6 Jan 2025

Response to Reviewer 1 Comments

Thank you very much for taking the time to review this manuscript. According to your suggestions, we have made a substantial revision of the paper from several aspects, such as Mathematical Formulas, Regression Results, and Conclusions, so as to get your approval. And we marked the modified content in yellow. If there is any content that needs to be improved, please contact us.

Comments 1: The discussion of the mediating variable is unclear. The mediating variable cannot be seen from the regression equation. Risk-taking and debt financing are more like two independent dependent variables. So what is the logic between the two? This is the biggest problem of this article.

Response 1: 1. With regard to mediating variable, the revised draft adds model (3)�and puts the second- generation ownership, risk-taking and debt financing into the model (3) for regression test of mediating effect.Please see Lines 200-205. In addition, the revised draft also uses the Bootstrap method to test the mediating effect. Please see Lines 338-349 and the Column (1) and (2) of Table 9.

Comments 2: In 3.1 Sample selection and data collection, the data selection year is not specified, and it can only be found in the paper conclusion, which needs to be explained clearly in 3.1. Why use the data of these 10 years? Because it is now 2024, the data is obviously a bit outdated.

Response 2: The original paper began writing in 2020, when the latest data available was 2019. In addition, the sample size of intergenerational inheritance of Chinese family enterprises before 2008 is very small, and the debt financing of enterprises in 2009 was affected by the 2008 financial crisis, so the data of the original paper chose 2010 to 2019. The revised draft updated the data from 2003 to 2023, and supplemented the reasons for the data selection in the 3.1 Sample selection and data collection of the revised draft.Please see Lines 146-147. In additional,The empirical part of the revised draft also uses new data for analysis.

Comments 3: Endogeneity problem. Using only lagged terms cannot completely avoid the endogeneity problem. The author needs to add an endogeneity test and select appropriate IV for verification.

Response 3: Thank you for pointing this out. Based on your suggestion, the revised draft adds endogenous test, which is verified by propensity matching score method and instrumental variable method. Please see Lines 290-317

Comments 4: In the empirical part, the institutional variable inhibits the relationship between the two, but from the data test results, its sign is positive, so from the mathematical logic, it should be a positive adjustment. Therefore, when defining the institutional variable, it is necessary to explain whether the larger the data, the better the institutional environment, or vice versa.

Response 4: In the empirical part, from the data test results, the symbol of the coefficient of the interaction between institutional environment and second-generation ownership is positive. Whether the regulatory effect of the institutional environment is positive or negative needs to be considered in combination with the relationship between second-generation ownership and debt financing in the previous benchmark regression. The coefficient of second-generation ownership in the previous benchmark regression is negative. Therefore, the better the institutional environment, the negative relationship between second-generation ownership and debt financing is weakened, or vice versa. Please see Lines 350-363.

Comments 5: In further research, the author distinguished the institutional environment. What is the necessity for this? The overall index of institutional environment has reflected the differences in different regions. In addition, some regression results are significant while others are not, which economic reasons here are not clearly explained.

Response 5: In further research, the paper distinguishes the specific elements of the institutional environment, because the score of the institutional environment is calculated by the scores of these specific environmental elements. The paper hopes to explore the specific impact of various environmental factors on the debt financing of family enterprises in the intergenerational inheritance period, in order to put forward more targeted suggestions. In addition, some regression results are significant, while others are not, and the possible reasons are given in the revised draft.Please see Lines 399-405.

---

## [Decision Letter · Decision Letter 1]

PONE-D-24-11863R1Ownership Succession, Risk Taking and Debt Financing from the Perspective of the Institutional EnvironmentPLOS ONE

Dear Dr. Tang,

Thank you for submitting your manuscript to PLOS ONE. After careful consideration, we feel that it has merit but does not fully meet PLOS ONE’s publication criteria as it currently stands. Therefore, we invite you to submit a revised version of the manuscript that addresses the points raised during the review process.

We look forward to receiving your revised manuscript.

Kind regards,

Gianluca Mattarocci, PhD

Academic Editor

PLOS ONE

Reviewers' comments:

Reviewer's Responses to Questions

**Comments to the Author**

1. If the authors have adequately addressed your comments raised in a previous round of review and you feel that this manuscript is now acceptable for publication, you may indicate that here to bypass the “Comments to the Author” section, enter your conflict of interest statement in the “Confidential to Editor” section, and submit your "Accept" recommendation.

Reviewer #1: All comments have been addressed

Reviewer #2: (No Response)

2. Is the manuscript technically sound, and do the data support the conclusions?

Reviewer #1: Yes

Reviewer #2: Yes

3. Has the statistical analysis been performed appropriately and rigorously? 

Reviewer #1: Yes

Reviewer #2: Yes

4. Have the authors made all data underlying the findings in their manuscript fully available?

Reviewer #1: Yes

Reviewer #2: Yes

5. Is the manuscript presented in an intelligible fashion and written in standard English?

Reviewer #1: Yes

Reviewer #2: No

6. Review Comments to the Author

Reviewer #1: The author's revisions were quite thorough and effectively resolved my concerns. The academic and scientific nature of the article has been greatly improved.

Reviewer #2: Overall, it is a very good study, however, following points can be considered for further development:

1. The research gap needs to be strengthened.

2. The introduction section needs to be strongly built.

2. The authors should use theory to build the framework of the study. Agency theory may be more relevant.

3. Hypotheses formulation needs to be logical in the light of theory used.

4. Why sample started from 2003? This needs to be justified.

5. The results need to be discussed in the light of theory and in the light of previous studies.

6. Theoretical and practical implications needs to be further strengthened.

7. PLOS authors have the option to publish the peer review history of their article (what does this mean? ). If published, this will include your full peer review and any attached files.

**Do you want your identity to be public for this peer review?** For information about this choice, including consent withdrawal, please see our Privacy Policy .

Reviewer #1: **Yes: ** Baili Yang

Reviewer #2: **Yes: ** Ali Amin

---

## [Author Response · Author response to Decision Letter 2]

28 Apr 2025

Reviewer #2: Overall, it is a very good study, however, following points can be considered for further development:

1. The research gap needs to be strengthened.

Answer The author has reorganized the literature on debt financing in intergenerational inheritance of family businesses, and compared in detail the viewpoints, methods, and conclusions of different previous studies, pointing out the shortcomings of previous research. Please refer to lines 39-64 for details

2. The introduction section needs to be strongly built.

Answer The author added institutional and practical backgrounds related to the research in the introduction, pointed out the shortcomings of previous research, and introduced research questions around agency theory. Please refer to lines 21-30, 39-64, and 65-69 for details.

2. The authors should use theory to build the framework of the study. Agency theory may be more relevant.

Answer The author has used agency theory to reconstruct the research framework�in the introduction section of the paper, agency theory is introduced to propose research questions. In the hypothesis proposal section, logical reasoning is carried out based on agency theory. In the conclusion and discussion section, the agency theory is used to discuss the research results. Please refer to lines 65-69, 89-117, 121-142, 161-182, and 465-505 for details.

3. Hypotheses formulation needs to be logical in the light of theory used.

Answer The author used agency theory to conduct a logical reasoning of the hypotheses again. Please refer to lines 89-117121-142 and 161-182 for details.

4. Why sample started from 2003? This needs to be justified.

Answer The author provided additional explanations on why sample started from 2003. Please refer to lines 191-198 for details.

5. The results need to be discussed in the light of theory and in the light of previous studies.

Answer The author has conducted a new discussion on the research conclusions based on agency theory and previous studies. Please refer to lines 465-482 for details.

6. Theoretical and practical implications needs to be further strengthened.

Answer The author provided a new exposition on the theoretical and practical significance at the end of the paper. Please refer to lines 483-505 for details.

---

## [Decision Letter · Decision Letter 2]

Ownership Succession, Risk Taking and Debt Financing from the Perspective of the Institutional Environment

PONE-D-24-11863R2

Dear Dr. Tang,

We’re pleased to inform you that your manuscript has been judged scientifically suitable for publication and will be formally accepted for publication once it meets all outstanding technical requirements.

Kind regards,

Stefan Cristian Gherghina, PhD. Habil.

Academic Editor

PLOS ONE

Additional Editor Comments (optional):

Reviewers' comments:

Reviewer's Responses to Questions

**Comments to the Author**

1. If the authors have adequately addressed your comments raised in a previous round of review and you feel that this manuscript is now acceptable for publication, you may indicate that here to bypass the “Comments to the Author” section, enter your conflict of interest statement in the “Confidential to Editor” section, and submit your "Accept" recommendation.

Reviewer #1: All comments have been addressed

Reviewer #2: All comments have been addressed

2. Is the manuscript technically sound, and do the data support the conclusions?

Reviewer #1: (No Response)

Reviewer #2: Yes

3. Has the statistical analysis been performed appropriately and rigorously? 

Reviewer #1: (No Response)

Reviewer #2: Yes

4. Have the authors made all data underlying the findings in their manuscript fully available?

Reviewer #1: (No Response)

Reviewer #2: Yes

5. Is the manuscript presented in an intelligible fashion and written in standard English?

Reviewer #1: (No Response)

Reviewer #2: Yes

6. Review Comments to the Author

Reviewer #1: (No Response)

Reviewer #2: Dear Authors

Thank you very much for incorporating my suggestions. I am satisfied and recommend acceptance of this paper.

7. PLOS authors have the option to publish the peer review history of their article (what does this mean? ). If published, this will include your full peer review and any attached files.

**Do you want your identity to be public for this peer review?** For information about this choice, including consent withdrawal, please see our Privacy Policy .

Reviewer #1: No

Reviewer #2: **Yes: ** Ali Amin

---

## [Editor Report · Acceptance letter]

PONE-D-24-11863R2

PLOS ONE

Dear Dr. Tang,

I'm pleased to inform you that your manuscript has been deemed suitable for publication in PLOS ONE. Congratulations! Your manuscript is now being handed over to our production team.

Kind regards,

on behalf of

Dr. Stefan Cristian Gherghina

Academic Editor

PLOS ONE